# Causal Discovery from Soft Interventions with Unknown Targets: Characterization and Learning

**Amin Jaber**
Department of Computer Science
Purdue University, USA
jaber0@purdue.edu

**Murat Kocaoglu**
MIT-IBM Watson AI Lab
IBM Research MA, USA
murat@ibm.com

**Karthikeyan Shanmugam**
MIT-IBM Watson AI Lab
IBM Research NY, USA
karthikeyan.shanmugam2@ibm.com

**Elias Bareinboim**
Department of Computer Science
Columbia University, USA
eb@cs.columbia.edu

## Abstract

One fundamental problem in the empirical sciences is of reconstructing the causal structure that underlies a phenomenon of interest through observation and experimentation. While there exists a plethora of methods capable of learning the equivalence class of causal structures that are compatible with observations, it is less well-understood how to systematically combine observations and experiments to reconstruct the underlying structure. In this paper, we investigate the task of structural learning in non-Markovian systems (i.e., when latent variables affect more than one observable) from a combination of observational and soft experimental data when the interventional targets are unknown. Using causal invariances found across the collection of observational and interventional distributions (not only conditional independences), we define a property called $\Psi$-Markov that connects these distributions to a pair consisting of (1) a causal graph $\mathcal{D}$ and (2) a set of interventional targets $\mathcal{I}$. Building on this property, our main contributions are two-fold: First, we provide a graphical characterization that allows one to test whether two causal graphs with possibly different sets of interventional targets belong to the same $\Psi$-Markov equivalence class. Second, we develop an algorithm capable of harnessing the collection of data to learn the corresponding equivalence class. We then prove that this algorithm is sound and complete, in the sense that it is the most informative in the sample limit, i.e., it discovers as many tails and arrowheads as can be oriented within a $\Psi$-Markov equivalence class.

## 1 Introduction

Learning cause-and-effect relationships is one of the fundamental problems for various fields, including biology [28, 6], epidemiology [26], and economics [12]. A prominent approach for causal discovery models the underlying system as a causal graph represented by a directed acyclic graph (DAG), where nodes denote random variables (measured or latent) and directed edges denote causal effects from tails to arrowheads [22, 29, 24]. Accordingly, the task of structural learning entails piecing together the constraints found in the data (and implied by the underlying, unobserved causal system) to infer the corresponding causal graph. In practice, however, these constraints are almost never sufficient to determine the true causal graph, and a collection of compatible graphs ends up being the target of the analysis, which forms what is known as an *equivalence class* (EC).

The formal understanding and characterization of equivalence classes have been an important part of the causal discovery literature for various reasons. For instance, one needs to understand how the output of a learning algorithm relates to the true underlying system that they are trying to infer. Also, ECs are defined with respect to certain constraints implied by the underlying structure in the data (e.g., conditional independences (CIs)), which need to be made explicit and fully understood if one wants to learn from the data (including due to false positives, negatives). Whenever only observational (non-experimental) data is available, the *Markov equivalence class* (for short, MEC) characterizes the causal graphs that imply, by the d-separation criterion, the same set of conditional independences (CIs) over the measured variables [32]. The availability of interventional (i.e., experimental) data opens up new opportunities to reduce the size of the equivalence class down, possibly to recover the true causal graph [10, 18, 8]. An intervention on a (measured) variable $X$ modifies the mechanism by which it is generated, inducing an interventional distribution over the measured variables $\mathbf{V}$, denoted as $P_X(\mathbf{V})$ or $P_X$ [22]. The works in [9, 34, 17] characterize the so called $\mathcal{I}$-Markov equivalence class, which uses distributional invariances within and across the available mixture of observational and interventional distributions. For instance, the graphs $\mathcal{D}_1 = \{X \to Y, X \leftarrow L \to Y\}$, where $L$ is latent, and $\mathcal{D}_2 = \{X \leftarrow Y\}$ are indistinguishable from observational data alone as no CI is implied (i.e., $X \not\perp\!\!\!\perp Y$). Still, they are immediately distinguishable given $\langle P, P_X \rangle$ by contrasting $P(Y)$ and $P_X(Y)$.

In this paper, we investigate *soft interventions* such that the mechanism of an intervened node $V_i$ is modified without fully eliminating the effect of its parents. This operation is also known as a mechanism change [31] or a parameter change [5], and it presents in many settings a more realistic model than *hard* or *perfect* interventions, where variables are forced to a fixed value (see also [2, 3, 33, 22, Sec. 3.2.2]). Furthermore, we relax the interventional setting by assuming the targets of the intervention to be unknown. For example, in molecular biology, the effects of various added chemicals to the cell are not set to one specific value nor they are precisely known [4].

The unknown interventional target setting requires a separate treatment than the known one since it's certainly less informative, i.e., the equivalence class of causal graphs is usually larger (never smaller), and many of the proposed characterizations and algorithms do not immediately apply. For concreteness, consider the two causal graphs mentioned above ($\mathcal{D}_1, \mathcal{D}_2$) that are distinguishable under a known interventional target set $\mathcal{I} = \langle \emptyset, \{X\} \rangle$, where $\emptyset$ denotes the observational setting and $\{X\}$ denotes an intervention on the variable. However, they turn out to be indistinguishable when $\mathcal{D}_1$ is associated with $\mathcal{I}_1 = \langle \emptyset, \{X\} \rangle$ but $\mathcal{D}_2$ is associated with $\mathcal{I}_2 = \langle \emptyset, \{X, Y\} \rangle$. In other words, the distributional invariances (to be formally defined in Section 3) accept both hypotheses that a pair of distributions with unknown intervention targets $\langle P_1, P_2 \rangle$ is generated by $\langle \mathcal{D}_1, \mathcal{I}_1 \rangle$ or $\langle \mathcal{D}_2, \mathcal{I}_2 \rangle$, where $P_1 = P(\mathbf{V})$ in both, $P_2 = P_X(\mathbf{V})$ for $\mathcal{D}_1$, and $P_2 = P_{X,Y}(\mathbf{V})$ for $\mathcal{D}_2$. Since the data is compatible with both $\mathcal{I}_1, \mathcal{I}_2$ for different graphs, the EC is underdetermined relative to known interventional targets.

Various approaches have been proposed to learn the causal graph from interventional distributions with unknown interventional targets. The works in [4, 23, 38, 30, 15] assume Markovianity (the absence of latent confounders). Another approach described in [27] learns cyclic causal graphs assuming linearity from unknown shift interventions, which is a specific type of soft interventions. Finally, [20] presents a framework called JCI, which pools the various distributions together by constructing context variables and then running traditional learning algorithms to identify the corresponding EC.[1]

In this work, we take a more fundamental approach and explicitly formalize the constraints that are being tested among the mixture of observational (when available) and interventional distributions, as well as provide a characterization of the equivalence class with respect to those constraints. Assuming a tuple of distributions $\mathbf{P} = \langle P_i \rangle_{i=1}^m$ is generated by the same system, i.e., causal graph with latents, we define a property called $\Psi$-Markov that connects $\mathbf{P}$ to a pair consisting of (1) a causal graph $\mathcal{D}$ and (2) a set of interventional targets $\mathcal{I}$. Building on this property in Section 3, we provide a graphical characterization that allows one to test whether two causal graphs with possibly different sets of intervention targets belong to the same $\Psi$-Markov equivalence class. We show that a graphical characterization for the causally sufficient case follows as a special case of this result. In Section 4, we develop $\Psi$-FCI, a constraint-based algorithm capable of harnessing the distributional invariances found across the combined data to learn the corresponding equivalence class. Finally, we prove that this algorithm is sound and complete, in the sense that it is the most informative in the sample limit. In other words, $\Psi$-FCI discovers as many tails and arrowheads as can be oriented within the corresponding $\Psi$-Markov equivalence class. In summary, our contributions are as follow:

1. We formulate a graphical characterization to test whether two pairs of causal graphs and their corresponding interventional target sets, $\langle \mathcal{D}_1, \mathcal{I}_1 \rangle$ and $\langle \mathcal{D}_2, \mathcal{I}_2 \rangle$, are in the same $\Psi$-Markov equivalence class, i.e., they are indistinguishable with respect to the available datasets.

2. We develop a sound and complete algorithm to learn equivalence classes of causal graphs from a collection of observational and experimental distributions with unknown interventional targets.

## 2 Preliminaries

We introduce in this section the necessary concepts and notation used throughout the paper. Upper case letters denote random variables and lower case letters denote an assignment. Also, bold letters denote sets. For $\mathbf{X}, \mathbf{Y}, \mathbf{Z}$, the CI relation $\mathbf{X}$ *is independent of* $\mathbf{Y}$ *conditioned on* $\mathbf{Z}$ is written as $\mathbf{X} \perp\!\!\!\perp \mathbf{Y} | \mathbf{Z}$. The d-separation statement $\mathbf{X}$ *is d-separated from* $\mathbf{Y}$ *given* $\mathbf{Z}$ *in graph* $\mathcal{D}$ is written as $(\mathbf{X} \perp\!\!\!\perp \mathbf{Y} | \mathbf{Z})_{\mathcal{D}}$. $\mathcal{D}_{\overline{\mathbf{X}}}$ denotes the graph obtained from $\mathcal{D}$ where all the incoming edges to the nodes in $\mathbf{X}$ are removed. Similarly, $\mathcal{D}_{\underline{\mathbf{X}}}$ denotes the removal of outgoing edges. We assume there is no selection bias. A star on edge endpoints is used as a wildcard to denote circle, arrowhead, or tail.

**Causal Bayesian Network (CBN):** Let $P(\mathbf{V})$ be a probability distribution over a set of variables $\mathbf{V}$, and $P_{\mathbf{x}}(\mathbf{V})$ denote the distribution resulting from the *hard intervention do*$(\mathbf{X} = \mathbf{x})$, which sets $\mathbf{X} \subseteq \mathbf{V}$ to constants $\mathbf{x}$. Let $\mathbf{P}^*$ denote the set of all interventional distributions $P_{\mathbf{x}}(\mathbf{V})$, for all $\mathbf{X} \subseteq \mathbf{V}$, including $P(\mathbf{V})$. A directed acyclic graph (DAG) over $\mathbf{V}$ is said to be a *causal Bayesian network* compatible with $\mathbf{P}^*$ if and only if, for all $\mathbf{X} \subseteq \mathbf{V}$, $P_{\mathbf{x}}(\mathbf{v}) = \prod_{\{i | V_i \notin \mathbf{X}\}} P(v_i | \mathbf{pa}_i)$, for all $\mathbf{v}$ consistent with $\mathbf{x}$, and where $\mathbf{Pa}_i$ is the set of parents of $V_i$ [22, 1, pp. 24]. Given that a subset of the variables are unmeasured or latent, $\mathcal{D}(\mathbf{V} \cup \mathbf{L}, \mathbf{E})$ will represent the causal graph where $\mathbf{V}$ and $\mathbf{L}$ denote the measured and latent variables, respectively, and $\mathbf{E}$ denotes the edges. Following the convention in [22], for simplicity, a dashed bi-directed edge is used instead of the corresponding latent variables. The CI relations can be read from the graph using a graphical criterion known as *d-separation* [21].

**Soft Interventions:** Under this type of interventions, the original conditional distributions of the intervened variables $\mathbf{X}$ are replaced with new ones, without completely eliminating the causal effect of the parents. Accordingly, the interventional distribution $P_{\mathbf{X}}(\mathbf{v})$ for $\mathbf{X} \subseteq \mathbf{V}$ is such that $P^*(X_i | Pa_i) \neq P(X_i | Pa_i)$, $\forall X_i \in \mathbf{X}$. We refer to the mixture of observational and interventional distributions as interventional for simplicity, which factorizes as follow:

$$P_{\mathbf{X}}(\mathbf{v}) = \sum_{\mathbf{L}} \prod_{\{i | X_i \in \mathbf{X}\}} P^*(x_i | \mathbf{pa}_i) \prod_{\{j | T_j \notin \mathbf{X}\}} P(t_j | \mathbf{pa}_j) \tag{1}$$

**Ancestral Graphs:** A *mixed* graph can contain directed and bi-directed edges. $A$ is an ancestor of $B$ if there is a directed path from $A$ to $B$. $A$ is a *spouse* of $B$ if $A \leftrightarrow B$ is present. If $A$ is both a spouse and an ancestor of $B$, this creates an *almost directed cycle*. A path is a sequence of edges joining a unique sequence of nodes. An *inducing path* relative to $\mathbf{L}$ is a path on which every non-endpoint node $X \notin \mathbf{L}$ is a collider on the path (i.e., both edges incident to the node are into it) and every collider is an ancestor of an endpoint of the path. A mixed graph is *ancestral* if it does not contain directed or almost directed cycles. It is *maximal* if there is no inducing path (relative to the empty set) between any two non-adjacent nodes. A *Maximal Ancestral Graph* (MAG) is a graph that is both ancestral and maximal [25]. Given a causal graph $\mathcal{D}(\mathbf{V} \cup \mathbf{L}, \mathbf{E})$, a unique MAG $\mathcal{M}_{\mathcal{D}}$ over $\mathbf{V}$ can be constructed such that both the independence and the ancestral relations among $\mathbf{V}$ are retained; see, [36, p. 6].

A triple $\langle X, Y, Z \rangle$ is an unshielded triple if $X$ and $Y$ are adjacent, $Y$ and $Z$ are adjacent, and $X$ and $Z$ are not adjacent. If both edges are into $Y$, then the triple is referred to as *unshielded collider*. A path between $X$ and $Y$, $p = \langle X, \ldots, W, Z, Y \rangle$, is *discriminating* for $Z$ if every node between $X$ and $Z$ is a collider on $p$ and is a parent of $Y$. Two MAGs are Markov equivalent if and only if (1) they have the same adjacencies; (2) the same unshielded colliders; and (3) if a path $p$ is a discriminating path for $Z$ in both graphs, then $Z$ is a collider on $p$ in one graph if and only if it is a collider on $p$ in the other. A *PAG* represents an MEC of a MAG and is learnable from data. The output of the celebrated FCI algorithm is a PAG, which is proven sound and complete for the corresponding MEC [37].

## 3 Interventional Equivalence with Unknown Targets

In this section, we formalize the notion of interventional equivalence class when the interventional targets are unknown. Let $V_i^j$ denote an intervention on $V_i$ with a unique mechanism identified by

$j$. Hence, interventions denoted by $V_i^j$ and $V_i^k$ force different mechanisms such that $P_{V_i^j}(V_i|Pa_i) \neq P_{V_i^k}(V_i|Pa_i)$. Accordingly, each interventional target $\mathbf{I} = \{V_i^j\}_{i \in |\mathbf{V}|}$ is defined by a set of variables with corresponding mechanism identifiers denoted by $j \in \mathbb{N}$. We drop the mechanism identifier whenever it is not necessary. Next, we define an important operation between two interventional targets.

**Definition 1** (Symmetrical Difference $\Delta$). *Given two interventional targets $\mathbf{I}$ and $\mathbf{J}$, let $\mathbf{I}\Delta\mathbf{J}$ denote the symmetrical difference set such that $V_i \in \mathbf{I}\Delta\mathbf{J}$ if $V_i^j \in \mathbf{I}$ and $V_i^j \notin \mathbf{J}$ or vice versa.*

In words, the operation identifies the set of variables that have a unique interventional mechanism across two interventional targets. For example, given $\mathbf{I} = \{X^1, Y, Z\}$ and $\mathbf{J} = \{X^2, Y\}$, then $\mathbf{I}\Delta\mathbf{J} = \{X, Z\}$.

Next, we generalize the interventional Markov property ($\mathcal{I}$-Markov) [17] for the case when the targets are unknown, which we call $\Psi$-Markov. This property features prominently the different tests that emerge when a combination of observational and experimental distributions is available.

**Definition 2** ($\Psi$-Markov Property). *Let $\mathcal{D} = (\mathbf{V} \cup \mathbf{L}, \mathbf{E})$ denote a causal graph, let $\mathbf{P}$ denote an ordered tuple of distributions, and let $\mathcal{I}$ denote an ordered tuple of interventional targets such that $|\mathbf{P}| = |\mathcal{I}|$. Tuple $\mathbf{P}$ satisfies the $\Psi$-Markov property with respect to the pair $\langle \mathcal{D}, \mathcal{I} \rangle$ if the following holds for disjoint $\mathbf{Y}, \mathbf{Z}, \mathbf{W} \subseteq \mathbf{V}$:*

*(a) For $\mathbf{I}_i \in \mathcal{I}$:*    $P_i(\mathbf{y}|\mathbf{w}, \mathbf{z}) = P_i(\mathbf{y}|\mathbf{w})$    *if $\mathbf{Y} \perp\!\!\!\perp \mathbf{Z}|\mathbf{W}$ in $\mathcal{D}$*

*(b) For $\mathbf{I}_i, \mathbf{I}_j \in \mathcal{I}$:*    $P_i(\mathbf{y}|\mathbf{w}) = P_j(\mathbf{y}|\mathbf{w})$    *if $\mathbf{Y} \perp\!\!\!\perp \mathbf{K}|\mathbf{W} \setminus \mathbf{W_K}$ in $\mathcal{D}_{\underline{\mathbf{W_K}}, \overline{\mathbf{R(W)}}}$,*

*where $\mathbf{K} := \mathbf{I}_i\Delta\mathbf{I}_j$, $\mathbf{W_K} =: \mathbf{W} \cap \mathbf{K}$, $\mathbf{R} := \mathbf{K}\backslash\mathbf{W_K}$, and $\mathbf{R(W)} \subseteq \mathbf{R}$ are non-ancestors of $\mathbf{W}$ in $\mathcal{D}$.*

*$\Psi_{\mathcal{I}}(\mathcal{D})$ denotes set of distribution tuples that satisfy the $\Psi$-Markov property with respect to $\langle \mathcal{D}, \mathcal{I} \rangle$.*

For concreteness and to illustrate this definition, we provide two examples with tuples of distributions that satisfy and do not satisfy the corresponding $\Psi$-Markov property, respectively.

**Example 1.** *Consider the causal graph $\mathcal{D}^* = \{X \rightarrow Y, X \leftarrow L \rightarrow Y\}$ where $L$ is a latent node, and let the pair of distributions $\langle P_1, P_2 \rangle$ be the result of intervening on the targets $\mathcal{I}^* = \langle \emptyset, \{X\}\rangle$. It is easy to check that $\mathbf{P}$ satisfies the $\Psi$-Markov property with respect to $\langle \mathcal{D}^*, \mathcal{I}^*\rangle$ as no constraint of type (a) or (b) is applicable. For example, if $(Y \perp\!\!\!\perp X)_{\mathcal{D}^*_{\underline{X}}}$, then the invariance $P_1(y|x) = P_2(y|x)$ must hold. Since the d-separation fails, the invariance is not required. Similarly, $\mathbf{P}$ satisfies the $\Psi$-Markov property with respect to $\langle \mathcal{D}, \mathcal{I} \rangle$ where $\mathcal{D} = \{X \leftarrow Y\}$ and $\mathcal{I} = \langle \emptyset, \{X, Y\}\rangle$ or $\mathcal{I} = \langle\{X\}, \{Y\}\rangle$.*

**Example 2.** *Consider the pair $\langle \mathcal{D}^*, \mathcal{I}^*\rangle$ and the corresponding tuple of distributions $\mathbf{P}$ from Ex. 1. We check if $\mathbf{P}$ satisfies the $\Psi$-Markov property with respect to $\langle \mathcal{D}^*, \mathcal{I}\rangle$ for $\mathcal{I} = \langle \emptyset, \{Y\}\rangle$. Now, $\mathbf{K} = \emptyset\Delta\{Y\} = \{Y\}$ and we have $(X \perp\!\!\!\perp Y)_{\mathcal{D}_{\overline{Y}}}$, so $P_1(X) = P_2(X)$ should hold according to Constraint (b). However, the invariance does not hold simply because $P_2$, in truth, corresponds to the interventional distribution on X. Therefore, $\mathbf{P}$ does not satisfy the $\Psi$-Markov property with respect to $\langle \mathcal{D}^*, \mathcal{I}\rangle$.*

A few remarks are relevant about the $\Psi$-Markov property at this point. First, an ordered tuple of interventional distributions $\mathbf{P}$ with unknown interventional targets is said to satisfy the $\Psi$-Markov property if two qualitatively different types of constraints hold – (a) the "traditional" Markov property, where separation in the causal graph $\mathcal{D}$ implies CI in the corresponding distribution (including the interventional ones); (b) invariances across pairs of distributions given separation statements in the mutilated graph. These mutilations depend on the symmetrical difference set ($\mathbf{K}$) of the interventional targets. Intuitively, should $\mathbf{I}_i, \mathbf{I}_j$ correspond to the true interventional targets of $P_i, P_j$, respectively, (b) verifies distributional invariances between $P_{\mathbf{I}_i}$ and $P_{\mathbf{I}_j}$ if the corresponding separation holds.

Second, the importance of the property stems from the fact that a tuple of interventional distributions generated by a causal graph $\mathcal{D}$ satisfies the $\Psi$-Markov property relative to it and the corresponding true interventional targets. See [13, Thm. 4 in Appx. A] for an explicit statement. Third, we note that if the interventional targets are known (i.e., $\mathcal{I}_1 = \mathcal{I}_2$), the $\Psi$-Markov property still generalizes $\mathcal{I}$-Markov [17] by relaxing the assumption of *controlled experiment setting*. In practice, it may be hard to ascertain that interventions over the same variable are performed exactly in the same way, which makes this more refined characterization potentially interesting even to when the interventional target is known. Finally, decoupling the distributions from the corresponding interventional targets is instrumental to formulate interventional equivalence when the targets are unknown as shown below.

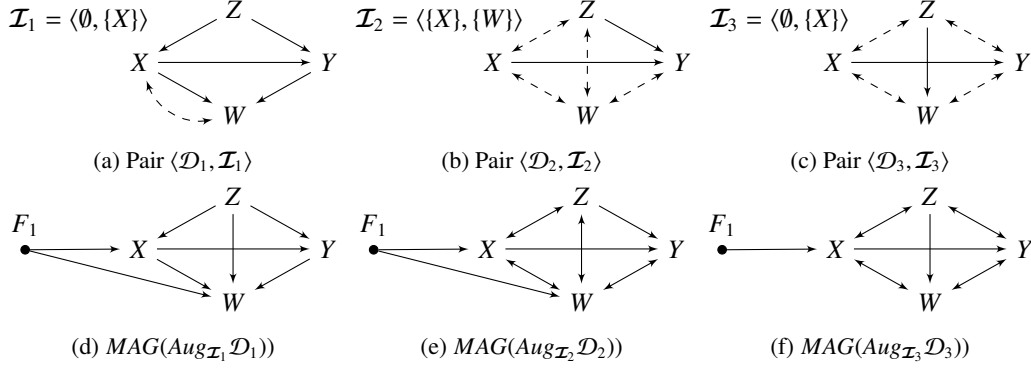

Figure 1: Pairs of causal graphs and intervention-target sets and the corresponding $\mathcal{I}$-MAGs.

**Definition 3** ($\Psi$-Markov Equivalence). *Given the causal graphs $\mathcal{D}_1 = (\mathbf{V} \cup \mathbf{L}_1, \mathbf{E}_1)$ and $\mathcal{D}_2 = (\mathbf{V} \cup \mathbf{L}_2, \mathbf{E}_2)$, and the corresponding interventional targets $\mathcal{I}_1, \mathcal{I}_2$, the pairs $\langle \mathcal{D}_1, \mathcal{I}_1 \rangle$ and $\langle \mathcal{D}_2, \mathcal{I}_2 \rangle$ are said to be $\Psi$-Markov equivalent if $\Psi_{\mathcal{I}_1}(\mathcal{D}_1) = \Psi_{\mathcal{I}_2}(\mathcal{D}_2)$.*

In words, two pairs of causal graphs and their corresponding sets of interventional targets $\langle \mathcal{D}_1, \mathcal{I}_1 \rangle$ and $\langle \mathcal{D}_2, \mathcal{I}_2 \rangle$ are $\Psi$-Markov equivalent if they can induce the same set of distribution tuples. In practice, it may be challenging to evaluate whether the premises of the $\Psi$-Markov property hold since they entail different graph mutilations of $\mathcal{D}$. In order to ameliorate this task, we build on the graph augmentation construction following [17].

**Definition 4** (Augmented graph). *Consider a causal graph $\mathcal{D} = (\mathbf{V} \cup \mathbf{L}, \mathbf{E})$ and a set of interventional targets $\mathcal{I}$. Let the multiset $\mathcal{K}$ be defined as such $\mathcal{K} = \{\mathbf{K}_1, \mathbf{K}_2, \ldots, \mathbf{K}_k\} = \{\mathbf{K} | \mathbf{I}, \mathbf{J} \in \mathcal{I} \wedge \mathbf{I} \Delta \mathbf{J} = \mathbf{K}\}$. The augmented graph of $\mathcal{D}$ with respect to $\mathcal{I}$, denoted as $Aug_{\mathcal{I}}(\mathcal{D})$, is the graph constructed as follows: $Aug_{\mathcal{I}}(\mathcal{D}) = (\mathbf{V} \cup \mathbf{L} \cup \mathcal{F}, \mathbf{E} \cup \mathcal{E})$ where $\mathcal{F} := \{F_i\}_{i \in [k]}$ and $\mathcal{E} = \{(F_i, j)\}_{i \in [k], j \in \mathbf{K}_i}$.*

For each pair of interventional targets $\mathbf{I}, \mathbf{J} \in \mathcal{I}$ such that $\mathbf{K} = \mathbf{I} \Delta \mathbf{J}$, the augmented graph appends the causal graph $\mathcal{D}$ with a utility F-node that is a parent to each node in $\mathbf{K}$. The significance of this construction follows from Proposition 1 where separation statements in the $\Psi$-Markov definition are tied (shown to be equivalent, formally speaking) to ones in the augmented graph, with no need to perform any graphical mutilation. The result is illustrated in the following example.

**Proposition 1.** *Consider a causal graph $\mathcal{D} = (\mathbf{V} \cup \mathbf{L}, \mathbf{E})$, a set of interventional targets $\mathcal{I}$, and the augmented graph $Aug_{\mathcal{I}}(\mathcal{D})$, where $\mathcal{F} = \{F_i\}_{i \in [k]}$. Let $\mathbf{K}_i$ be the set of nodes adjacent to $F_i$, $\forall i \in [k]$. The following equivalence relations hold for disjoint $\mathbf{Y}, \mathbf{Z}, \mathbf{W} \subseteq \mathbf{V}$, where $\mathbf{W}_i := \mathbf{W} \cap \mathbf{K}_i$, $\mathbf{R} := \mathbf{K}_i \setminus \mathbf{W}_i$.[2]*

$$(\mathbf{Y} \perp\!\!\!\perp \mathbf{Z} | \mathbf{W})_{\mathcal{D}} \iff (\mathbf{Y} \perp\!\!\!\perp \mathbf{Z} | \mathbf{W}, F_{[k]})_{Aug_{\mathcal{I}}(\mathcal{D})} \tag{2}$$

$$(\mathbf{Y} \perp\!\!\!\perp \mathbf{K}_i | \mathbf{W} \setminus \mathbf{W}_i)_{\mathcal{D}_{\underline{\mathbf{W}_i}, \overline{\mathbf{R(W)}}}} \iff (\mathbf{Y} \perp\!\!\!\perp F_i | \mathbf{W}, F_{[k] \setminus \{i\}})_{Aug_{\mathcal{I}}(\mathcal{D})} \tag{3}$$

**Example 3.** *Consider $\mathcal{D} = \{X \rightarrow Y \leftarrow L \rightarrow Z\}$ where $L$ is latent and let $\mathcal{I} = \langle \{X, Z^1\}, \{Z^2\} \rangle$. The corresponding augmented graph $\mathcal{D}'$ is composed of $\mathcal{D}$ appended with $X \leftarrow F_1 \rightarrow Z$. By Prop. 1, $(X \perp\!\!\!\perp Z)_{\mathcal{D}}$ can be tested by $(X \perp\!\!\!\perp Z | F_1)_{\mathcal{D}'}$. Also, $(Y \perp\!\!\!\perp \{X, Z\})_{\mathcal{D}_{\underline{X}, \overline{Z}}}$ can be tested as $(F_1 \perp\!\!\!\perp Y | X)_{\mathcal{D}'}$.*

Maximal Ancestral Graphs (MAGs) provide a convenient representation capable of preserving all the tested constraints in augmented graphs represented by d-separations [25]; see also [36, p. 6]. This is formalized in Definition 5 and the construct is referred to as an $\mathcal{I}$-MAG; see Example 4 below.

**Definition 5** ($\mathcal{I}$-MAG). *Given a causal graph $\mathcal{D} = (\mathbf{V} \cup \mathbf{L}, \mathbf{E})$ and a set of interventional targets $\mathcal{I}$, an $\mathcal{I}$-MAG is the MAG constructed over $\mathbf{V}$ from $Aug_{\mathcal{I}}(\mathcal{D})$, i.e., $MAG(Aug_{\mathcal{I}}(\mathcal{D}))$.*

**Example 4.** *Consider $\mathcal{D}^*$ from Ex. 1. $Aug_{\mathcal{I}}(\mathcal{D}^*) = \{F_1 \rightarrow X \rightarrow Y, X \leftarrow L \rightarrow Y\}$ for $\mathcal{I} = \langle \emptyset, \{X\} \rangle$. Then, the corresponding $\mathcal{I}$-MAG is $MAG(Aug_{\mathcal{I}}(\mathcal{D}^*)) = \{X \leftarrow F_1 \rightarrow Y, X \rightarrow Y\}$.*

Putting these results together, we derive next a graphical characterization for two causal graphs with corresponding sets of interventional targets to be $\Psi$-Markov equivalent.

**Theorem 1** (Ψ-Markov Characterization). *Given causal graphs $\mathcal{D}_1 = (\mathbf{V} \cup \mathbf{L}_1, \mathbf{E}_2), \mathcal{D}_2 = (\mathbf{V} \cup \mathbf{L}_2, \mathbf{E}_2)$ and corresponding sets of interventional targets $\mathcal{I}_1, \mathcal{I}_2, \langle \mathcal{D}_1, \mathcal{I}_1 \rangle$ and $\langle \mathcal{D}_2, \mathcal{I}_2 \rangle$ are Ψ-Markov equivalent if and only if for $\mathcal{M}_1 = MAG(Aug_{\mathcal{I}_1}(\mathcal{D}_1))$ and $\mathcal{M}_2 = MAG(Aug_{\mathcal{I}_2}(\mathcal{D}_2))$:[3]*

1. *$\mathcal{M}_1$ and $\mathcal{M}_2$ have the same skeleton;*

2. *$\mathcal{M}_1$ and $\mathcal{M}_2$ have the same unshielded colliders;*

3. *If a path $p$ is a discriminating path for a node $Y$ in both $\mathcal{M}_1$ and $\mathcal{M}_2$, then $Y$ is a collider on the path in one graph if and only if it is a collider on the path in the other.*

Theorem 1 states that the pairs $\langle \mathcal{D}_1, \mathcal{I}_1 \rangle$ and $\langle \mathcal{D}_2, \mathcal{I}_2 \rangle$ are Ψ-Markov equivalent if their corresponding $\mathcal{I}$-MAGs satisfy the corresponding three conditions, as illustrated in the example below.

**Example 5.** *Consider the pairs $\langle \mathcal{D}_1, \mathcal{I}_1 \rangle$ and $\langle \mathcal{D}_2, \mathcal{I}_2 \rangle$ in Figs. 1a and 1b, respectively. The corresponding $\mathcal{I}$-MAGs are shown in Figs. 1d and 1e satisfy the three conditions in Thm. 1, hence the pairs are Ψ-Markov equivalent. Note $\mathcal{K}$, according to Def. 4, is $\{\{X\}\}$ for $\mathcal{I}_1$ and $\{\{X, W\}\}$ for $\mathcal{I}_2$. Hence, $F_1$ is adjacent to $\{X\}$ in $Aug_{\mathcal{I}_1}(\mathcal{D}_1)$ and $F_1$ is adjacent to $\{X, W\}$ in $Aug_{\mathcal{I}_2}(\mathcal{D}_2)$. However, $F_1$ is adjacent to $W$ in Fig. 1d due to the inducing path $\langle F_1, X, W \rangle$ in $Aug_{\mathcal{I}_1}(\mathcal{D}_1)$. On the other hand, $\langle \mathcal{D}_3, \mathcal{I}_3 \rangle$ in Fig. 1c is not Ψ-Markov equivalent to either one of the other pairs as we violate all three conditions of Thm. 1. For instance, the $\mathcal{I}$-MAGs in Figs. 1e and 1f do not share the same skeleton, $\langle F_1, X, W \rangle$ is an unshielded collider only in Fig. 1f, and $p = \langle F_1, X, Z, Y \rangle$ is a discriminating path for $Z$ in both; however, $Z$ is a collider along $p$ in Fig. 1f while it is a non-collider in Fig. 1e.*

In a setting where the observational distribution is available and identified among the available distributions, it becomes necessary to fix $\emptyset$ across $\mathcal{I}_1, \mathcal{I}_2$, which is a special case of Thm. 1. Further, note that the graphical characterization introduced in [17] for causal graphs with known interventional targets is a special case of Thm. 1 whenever $\mathcal{I}_1 = \mathcal{I}_2$ with the controlled experiment setting.

## 3.1 Markovian Case

One special class of causal graphs that is of high interest in the literature is known as *Markovian*, where there is no latent variable affecting more than one observable node (i.e., no bidirected arrows). It follows from Theorem 1 the following graphical characterization for this class of models.

**Corollary 1.** *Given causal graphs without latents, $\mathcal{D}_1 = (\mathbf{V}, \mathbf{E}_2), \mathcal{D}_2 = (\mathbf{V}, \mathbf{E}_2)$, and the corresponding interventional targets $\mathcal{I}_1, \mathcal{I}_2$, the pairs $\langle \mathcal{D}_1, \mathcal{I}_1 \rangle$ and $\langle \mathcal{D}_2, \mathcal{I}_2 \rangle$ are Ψ-Markov equivalent if and only if $Aug_{\mathcal{I}_1}(\mathcal{D}_1)$ and $Aug_{\mathcal{I}_2}(\mathcal{D}_2)$ have (1) the same skeleton and (2) the same unshielded colliders.*

Note that under known interventional targets (i.e., $\mathcal{I}_1 = \mathcal{I}_2$), Corol. 1 recovers and generalizes the characterization in [34, Thm. 3.9] by encoding different interventional mechanisms and thus identifying a smaller equivalence class. For a more detailed comparison, we refer readers to Appendix D.1.

## 4 Learning Algorithm: Soundness and Completeness

We investigate in this section the problem of how to learn the Ψ-Markov EC (Def. 3) from a tuple of interventional distributions generated by some unknown pair $\langle \mathcal{D}, \mathcal{I} \rangle$. The characterization provided in Thm. 1 together with PAGs motivate the following definition of Ψ-*PAG*.

**Definition 6** (Ψ-PAG). *Given a pair of causal graph and interventional target, $\langle \mathcal{D}, \mathcal{I} \rangle$, let $\mathcal{M} = MAG(Aug_{\mathcal{I}}(\mathcal{D}))$, and let $[\mathcal{M}]$ be the set of $\mathcal{I}$-MAGs corresponding to all the pairs $\langle \mathcal{D}', \mathcal{I}' \rangle$ that are Ψ-Markov equivalent to $\langle \mathcal{D}, \mathcal{I} \rangle$. The Ψ-PAG for $\langle \mathcal{D}, \mathcal{I} \rangle$, denoted $\mathcal{P}$, is a graph such that:*

1. *$\mathcal{P}$ has the same adjacencies as $\mathcal{M}$, and any member of $[\mathcal{M}]$ does; and*

2. *every non-circle mark (tail or arrowhead) in $\mathcal{P}$ is an invariant mark in $[\mathcal{M}]$.*

Some remarks follow immediately from this definition. First, Ψ-PAG generalizes PAGs, as used in the observational case. Second, even though the augmented F-nodes are part of the Ψ-PAG, which is the very target of the learning process, they never transpire as random variables, and are

**Algorithm 1** $\Psi$-FCI: Algorithm for Learning a $\Psi$-PAG

---

    **Input:** Tuple of distributions $\mathbf{P} = \langle P_1, \ldots, P_m \rangle$
    **Output:** $\Psi$-PAG $\mathcal{P}$

1:  $\mathcal{F} \leftarrow \emptyset, k \leftarrow 0, \sigma : \mathbb{N} \to \mathbb{N} \times \mathbb{N}$
2:  **for** all pairs $P_i, P_j \in \mathbf{P}$ **do** $k \leftarrow k + 1, \mathcal{F} \leftarrow \mathcal{F} \cup \{F_k\}, \sigma(k) \to (i, j)$
3:  **Phase I: Skeleton**
4:  Form a complete graph $\mathcal{P}$ over $\mathbf{V} \cup \mathcal{F}$ with $\circ\!\!-\!\!\circ$ edges between every pair of nodes.
5:  **for** every pair $X, Y \in \mathbf{V} \cup \mathcal{F}$ **do**
6:     **if** $X \in \mathcal{F} \wedge Y \in \mathcal{F}$ **then** $\text{SepSet}(X, Y) \leftarrow \emptyset$, $\text{SepFlag} \leftarrow True$
7:     **else** $(\text{SepSet}(X, Y), \text{SepFlag}) \leftarrow \textsc{InvToSep}(\mathbf{P}, X, Y, \mathbf{V}, \mathcal{F}, \sigma)$
8:     **if** $\text{SepFlag} = True$ **then** Remove the edge between $X, Y$ in $\mathcal{P}$.
9:  **Phase II: Unshielded Colliders**
10:  $\mathcal{R}_0$: For every unshielded triple $\langle X, Z, Y \rangle$ in $\mathcal{P}$, orient it as $X * \!\!\rightarrow Z \leftarrow\!\! * Y$ iff $Z \notin \text{SepSet}(X, Y)$
11:  **Phase III: Orientation Rules**
12:  *Rule $\mathcal{R}^+$:* For any $F_k \in \mathcal{F}$, orient adjacent edges out of $F_k$.
13:  Apply the seven FCI rules in [37] ($\mathcal{R}_1 - \mathcal{R}_4, \mathcal{R}_8 - \mathcal{R}_{10}$) until none applies.

14:  **function** $\textsc{InvToSep}(\mathbf{P}, X, Y, \mathbf{V}, \mathcal{F}, \sigma)$
15:     $\text{SepSet} \leftarrow \emptyset$, $\text{SepFlag} \leftarrow False$
16:     **if** $X \notin \mathcal{F} \wedge Y \notin \mathcal{F}$ **then** Pick $P_i \in \mathbf{P}$ arbitrarily.
17:         **for** $\mathbf{W} \subseteq \mathbf{V} \setminus \mathcal{F}$ **do**
18:             **if** $P_i(y|\mathbf{w}, x) = P_i(y|\mathbf{w})$ **then** $\text{SepSet} \leftarrow \mathbf{W} \cup \mathcal{F}$, $\text{SepFlag} \leftarrow True$, **break**
19:     **else** Suppose $X \in \mathcal{F}, Y \notin \mathcal{F}$, and let $F_k$ denote $X$.
20:         $(i, j) \leftarrow \sigma(k)$
21:         **for** $\mathbf{W} \subseteq \mathbf{V} \setminus \{Y\}$ **do**
22:             **if** $P_i(y|\mathbf{w}) = P_j(y|\mathbf{w})$ **then** $\text{SepSet} \leftarrow \mathbf{W} \cup \mathcal{F} \setminus \{F_k\}$, $\text{SepFlag} \leftarrow True$, **break**
        **return** $(\text{SepSet}, \text{SepFlag})$

---

merely graphical instruments used to represent the equivalence class. In fact, the real invariance tests across distributions are stated in the $\Psi$-Markov property (Def. 2). Third, as expected in any learning setting, some type of faithfulness assumption is needed to infer graphical properties from the corresponding distributional constraints [37, 34, 17, 30]. Hence, we assume that the given collection of interventional distributions is *c-faithful* to the true generating causal graph $\mathcal{D}$ as defined next.

**Definition 7** (c-faithfulness). *Consider a causal graph $\mathcal{D}$. A tuple of distributions $\langle P_{\mathbf{I}} \rangle_{\mathbf{I} \in \mathcal{I}} \in \Psi_{\mathcal{I}}(\mathcal{D})$ is called c-faithful to $\mathcal{D}$ if the converse of each of the $\Psi$-Markov conditions (Def. 2) holds.*

The new algorithm is called $\Psi$-FCI and is shown in Alg. 1. $\Psi$-FCI starts by mapping every pair of distributions in $\mathbf{P}$ to a constructed F-node (line 2). In Phase I, $\Psi$-FCI learns the skeleton of the $\Psi$-PAG $\mathcal{P}$. It starts by creating a complete graph of circle edges ($\circ\!\!-\!\!\circ$) over $\mathbf{V} \cup \mathcal{F}$, and then uses the function $\textsc{InvToSep}(\cdot)$ at line 7, which we discuss next, to infer a separation set for every pair of nodes, if such a set exists. Line 6 handles a special case in which both nodes are F-nodes and are separable by the empty set, by construction. Phase II recovers the unshielded colliders $\langle X, Z, Y \rangle$ by checking that $Z$ does not belong to the corresponding separation set $\text{SepSet}(X, Y)$. Finally, the algorithm orients all the edges incident on F-nodes out of them in $\mathcal{R}^+$ followed by a subset of the FCI rules until none applies anymore. Note that we drop three of the FCI rules ($\mathcal{R}_5 - \mathcal{R}_7$) as they are only applicable in the presence of selection bias which we do not consider.

$\textsc{InvToSep}(\cdot)$ can be considered as the most fundamental part of $\Psi$-FCI. This function infers separation sets for pairs of nodes in $\mathcal{P}$ from the invariances found across the distributions.[4] The separation sets are key in $\Psi$-FCI to learn the skeleton and orient the edges of $\mathcal{P}$. If both $X$ and $Y$ are not F-nodes, then we pick an arbitrary distribution $P_i \in \mathbf{P}$ and check if there exists a subset of the variables $\mathbf{W}$ such that $(X \perp\!\!\!\perp Y | W)$ in $P_i$ (lines 3-5). The reason we choose an arbitrary distribution in $\mathbf{P}$ is that the set of conditional independences that can be read from an observational or interventional distribution is the same under soft interventions. For the next step, recall that every F-node is mapped to a unique pair of distributions in $\mathbf{P}$. If one of the two nodes is an F-node, denoted $F_k$, then we search for a subset of variables $\mathbf{W}$ such that $P_i(y|\mathbf{w}) = P_j(y|\mathbf{w})$ where $(i, j) \leftarrow \sigma(k)$. If such an invariance exists,

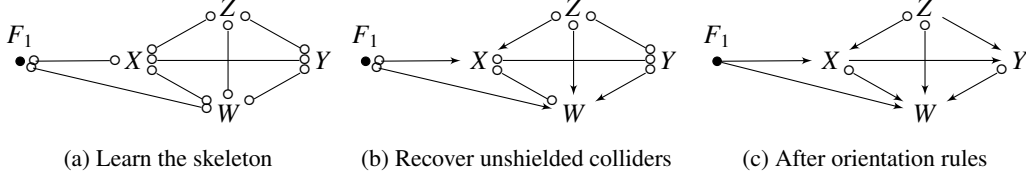

| (a) Learn the skeleton | (b) Recover unshielded colliders | (c) After orientation rules |

Figure 2: Different phases of Ψ-FCI to learn Ψ-PAG $\mathcal{P}$ given a tuple of distributions $\langle P_1, P_2 \rangle$ that is generated by the unknown pair $\langle \mathcal{D}, \mathcal{I} \rangle$, shown in Fig. 1a.

we mark $\mathbf{W} \cup \mathcal{F} \setminus \{F_k\}$ as a separating set between $F_k$ and $Y$. The validity of this function follows from the constraints of the Ψ-Markov property (Def. 2) and the equivalences in Proposition 1 coupled with the c-faithfulness assumption in Def. 7. We illustrate the use of Ψ-FCI in the example below.

**Example 6.** *Consider a tuple of distributions $\langle P_1, P_2 \rangle$ and let the pair $\langle \mathcal{D}, \mathcal{I} \rangle$ in Fig. 1a be the true and unknown causal graph and set of corresponding interventional targets. $\mathcal{I}$-MAG $\mathcal{M}$ is shown in Fig. 1d and the aim is to recover the corresponding Ψ-PAG $\mathcal{P}$. Fig. 2a shows the output of Phase I. For instance, $F_1$ and $Y$ are separable by $\{X, Z\}$, which is inferred by the distributional invariance $P_1(Y|X, Z) = P_2(Y|X, Z)$. Phase II recovers the unshielded colliders as shown in Fig. 2b. For example, $\langle F_1, W, Y \rangle$ is oriented as a collider since $W \notin \mathtt{SepSet}(F_1, Y) = \{X, Z\}$. Finally, Phase III applies the orientation rules which gives the graph in Fig. 2c. The edges incident on $F_1$ are oriented out of it by $\mathcal{R}^+$, $X \to Y$ by $\mathcal{R}_1$, the arrowhead on $X \circ\!\to W$ by $\mathcal{R}_2$, and $Z \to Y$ by $\mathcal{R}_4$.*

Putting these observations together, finally, the next theorem ascertains the soundness of Ψ-FCI.

**Theorem 2** (Ψ-FCI Soundness). *Assuming tuple $\mathbf{P}$ is generated by unknown pair $\langle \mathcal{D}, \mathcal{I} \rangle$, then Ψ-FCI is sound in the sample limit, i.e., $MAG(Aug_{\mathcal{I}}(\mathcal{D}))$ has the same skeleton as $\mathcal{P}_{\text{Ψ-FCI}}$, the Ψ-PAG learned by Ψ-FCI, and shares all its tail and arrowhead orientations.*

## 4.1 Ψ-FCI Completeness

One common question for any learning algorithm is how close it can get to the underlying causal structure. In the limit, one would like to discover all the invariant features of the corresponding Ψ-Markov EC, a property called completeness. Concretely, for every circle mark on an edge end in $\mathcal{P}_{\text{Ψ-FCI}}$, we need to establish the following. There exist two pairs $\langle \mathcal{D}_1, \mathcal{I}_1 \rangle$ and $\langle \mathcal{D}_2, \mathcal{I}_2 \rangle$ that are Ψ-Markov equivalent to the true pair $\langle \mathcal{D}, \mathcal{I} \rangle$ such that the corresponding $\mathcal{I}$-MAGs $\mathcal{M}_1$ and $\mathcal{M}_2$ have different marks for that end (i.e., one has a tail while the other has an arrowhead), as illustrated next.

**Example 7.** *Consider Ψ-PAG $\mathcal{P}$ in Fig. 2c from Ex. 6. $\mathcal{I}$-MAGs in Figs. 1d and 1e are both in the corresponding equivalence class represented by $\mathcal{P}$. Notice that for every circle mark in $\mathcal{P}$, the mark is a tail in one of the $\mathcal{I}$-MAGs while it is an arrowhead in the other. Hence, the orientations are complete. If the observational distribution is known, then consider the graph $\mathcal{D}_2$ in Fig. 1b with $\mathcal{I}_2^* = \langle \emptyset, \{X, W\} \rangle$. The corresponding $\mathcal{I}$-MAG is the one in Fig. 1e, so we obtain the same result.*

To understand the challenge of establishing Ψ-FCI's completeness, denote by $\mathcal{M}'$ a complete orientation of $\mathcal{P}_{\text{Ψ-FCI}}$. Note that even though $\mathcal{M}'$ may satisfy the three conditions of Thm. 1 with respect to the true $\mathcal{I}$-MAG $\mathcal{M}$, it is not implied that $\mathcal{M}'$ is a valid $\mathcal{I}$-MAG. Following the further requirements of Def. 5, we show next that there exists a pair $\langle \mathcal{D}', \mathcal{I}' \rangle$ such that $MAG(Aug_{\mathcal{I}'}(\mathcal{D}')) = \mathcal{M}'$.

**Lemma 1.** *Let $\mathcal{D}(\mathbf{V} \cup \mathbf{L}, \mathbf{E})$ denote a causal graph, $\mathcal{I}$ denote a set of interventional targets, $\mathcal{M} = MAG(Aug_{\mathcal{I}}(\mathcal{D}))$, and $\mathcal{M}'$ denote an arbitrary MAG over $\mathbf{V} \cup \mathcal{F}$. If the following holds:*

1. *All the edges incident on $\mathcal{F}$ in $\mathcal{M}'$ are out of $\mathcal{F}$; and,*

2. *$\mathcal{M}$ and $\mathcal{M}'$ share the same separation statements over $\mathbf{V} \cup \mathcal{F}$,*

*then there exists a pair $\langle \mathcal{D}', \mathcal{I}' \rangle$, including when $\emptyset \in \mathcal{I}$ and is fixed, such that $MAG(Aug_{\mathcal{I}'}(\mathcal{D}')) = \mathcal{M}'$. In other words, $\mathcal{M}'$ is an $\mathcal{I}$-MAG and the pair $\langle \mathcal{D}', \mathcal{I}' \rangle$ is Ψ-Markov equivalent to $\langle \mathcal{D}, \mathcal{I} \rangle$.*

Based on this result, completeness can be finally proved as shown next.

**Theorem 3** (Ψ-FCI Completeness). *Assuming tuple $\mathbf{P}$ is generated by unknown pair $\langle \mathcal{D}, \mathcal{I} \rangle$, then Ψ-FCI is complete, i.e., $\mathcal{P}$ contains all the common edge marks in the Ψ-Markov equivalence class.*

A few compelling connections emerge from this proposition. Leveraging Corollary 1, one can show that a variant of Ψ-FCI constrained to Meek's rules (which we called Ψ-PC) is also complete in the Markovian case for both known and unknown interventional targets. On the other hand, perhaps surprisingly, it can also be shown that the same result does not hold when sufficiency cannot be ascertained. For a more detailed discussion on these subtleties, see [13, Appendix C].

## 5 Conclusion

In this work, we investigated the problem of learning causal graphs with latent variables from a mixture of observational and interventional distributions with unknown interventional targets. We started by defining the Ψ-Markov property that connects a tuple of distributions with unknown targets to a pair of causal graph $\mathcal{D}$ and a corresponding possible interventional target set $\mathcal{I}$. Accordingly, two pairs $\langle \mathcal{D}_1, \mathcal{I}_1 \rangle$ and $\langle \mathcal{D}_2, \mathcal{I}_2 \rangle$ are said to be Ψ-Markov equivalent if they license the same tuples of distributions. Based on this refined equivalence relation, we derived a graphical characterization to evaluate whether two pairs are in the same Ψ-Markov equivalence class. Finally, we developed a sound and complete algorithm that recovers a Ψ-Markov equivalence class given a tuple of distributions. This work grounds the theoretical aspects of learning from unknown soft-interventions, thus, as we envision, paving the way for a new family of more robust and scalable methods that can address issues of computational and sample complexity, including score-based and approximation algorithms.

## Broader Impact

Learning cause-and-effect relationships is one of the fundamental problems for various fields, including biology [28, 6], epidemiology [26], and economics [12]. The introduced characterization and algorithm provide a clear understanding on how to accomplish this task while leveraging interventional data, even when the interventional targets are unknown. Moreover, the proposed approach can be instrumental towards explainability in artificial intelligence, which has been a topic of increasing importance recently. On the other hand, performing experiments to obtain interventional data poses some ethical challenges, such as randomizing the smoking factor which would require forcing individuals to smoke. Therefore, such limitations and concerns should be taken into consideration.

## Acknowledgments and Disclosure of Funding

Bareinboim and Jaber are supported in parts by grants from NSF IIS-1704352 and IIS-1750807 (CAREER). Kocaoglu and Shanmugam are supported by the MIT-IBM Watson AI Lab.

## Footnotes

[1] We provide detailed discussion on how some of these works compare to ours in the full report [13, Appx. D].

[2]All the proofs can be found in Appendices A & B of the full report [13].

[3]We assume the symmetrical difference sets $\mathcal{K}_1$, corresponding to $\mathcal{I}_1$, and $\mathcal{K}_2$, corresponding to $\mathcal{I}_2$, are indexed following the same pattern such that $\mathcal{K}_1 \ni \mathbf{K}_k = \mathbf{I}_i \Delta \mathbf{I}_j$ where $\mathbf{I}_i, \mathbf{I}_j \in \mathcal{I}_1$ iff $\mathcal{K}_2 \ni \mathbf{K}_k = \mathbf{I}_i \Delta \mathbf{I}_j$ where $\mathbf{I}_i, \mathbf{I}_j \in \mathcal{I}_2$. This is required to maintain the correspondence between the F-nodes in $\mathcal{M}_1$ and $\mathcal{M}_2$.

[4]There are different ways of implementing hypothesis testing for the distributional invariances, as required in line 22 of $\Psi$-FCI. In fact, these tests can be seen as evaluating statements in the form $|\hat{P}_i(y|\mathbf{w}) - \hat{P}_j(y|\mathbf{w})| \le \epsilon$, where the hat represents the empirical distribution. $\Psi$-FCI is agnostic to the particular implementation of the test, which is in general chosen based on the specific details of the setting.

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
