[Reviews · NeurIPS 2020]

Review 1

Summary and Contributions: In this paper, the authors extend the theoretical results in some previous works. The authors consider soft-intervention, unknown intervention targets, broken causal sufficiency. Specifically, the authors define \Psi-Markov equivalence for the (graph, intervention targets) pairs and provide the graphical criterion to test whether two pairs are equivalent. The authors give a sound and complete algorithm to discover the PAG given a set of distribution under different interventions.

Strengths: This paper provides solid theoretical results. To distinguish the (graph, intervention targets) pair, the authors define the corresponding equivalence class and provide the graphical characterization. The proposed algorithm is sound and complete. The paper is well written and mathematically sound.

Weaknesses: Compared to Kocaoglu et al. 2019, in which the authors provide a new perspective and an exciting solution for causal discovery without causal sufficiency by interventional data and observational data, more improvements could be expected in this paper.

Correctness: The claims appear to be correct. But I didn't check the proofs in detail.

Clarity: Yes.

Relation to Prior Work: Yes.

Reproducibility: Yes

Additional Feedback: C-faithfulness seems stricter than faithfulness assumption, which is shown prone to be broken when the variable number increases. Hence I am a bit concerning the possibility that the assumption is broken. I am sorry that I am not quite familiar with such region. Hence I am very likely to change my score if other reviewers or authors give persuasive opinions. ***********after the discussion********** Thanks for the rebuttal. After the discussion with other reviewers, I adjust my score accordingly.


Review 2

Summary and Contributions: Update: after the discussions, I agree with the other reviewers that the lack of comparison with [37] seems a bit concerning. Therefore, I decide to lower my score to weak accept. Still, I believe it's an interesting paper. I encourage the authors to provide a solid comparison with [37] and submit to the other venues if this did not go through. In many real world applications, people are usually faced with interventional data with unknown interventional targets. This calls for new methods that can successfully learn causal graphical graphical models with unknown-target interventional data. Learning causal graphical models with known-target interventional data is a well studied topic in this field, while there are still a lot of open questions for the task of causal discovery with unknown targets. In this paper, the authors provide a new graphical characterization of the equivalence classes of causal inference with unknown-target interventional data. A sound and complete algorithm is proposed to estimate the equivalent class.

Strengths: 1. Learning causal graphical models with observational and interventional data is an important topic in this field. While preliminary research mainly assume that all the interventional targets are known a priori, causal inference with unknown target interventional data is not well studied. This paper provides to my knowledge the first result to characterization the degree of identifiability from unknown-target interventional data, which facilities the researchers new techniques to develop methods to utilize such data for causal discovery; 2. The sound and complete algorithm looks inspiring. I especially like its comparisons with JCI in Section D.2, which shows clearly the improvement over the JCI framework. The JCI framework is well-known for its lack of completeness result. The completeness result from this paper has successfully filled this gap, together with an example showing clearly what has been improved over the JCI framework. 3. I did not check the proof in detail. But the proof seems correct to me and is technically sound. Overall I think this is an interesting paper and is relevant to the audience of the NeurIPS community.

Weaknesses: 1. In line 60, it should be “intervention on the variable X”.

Correctness: The proof seems to be correct and technically sound

Clarity: Yes

Relation to Prior Work: Very clear

Reproducibility: Yes

Additional Feedback:


Review 3

Summary and Contributions: This paper establishes a novel theoretical framework for reasoning about a causal graph and a set of unknown interventions with respect to observed interventional (and optionally an observational) distributions. The summary of the contributions is as follows: 1. Systematic development of the notion of Psi-Markov property and Psi-Equivalence Class. 2. The methodological tools needed to solve the learning problem (such as the augmented graph). 3. The learning algorithm, which builds on FCI. 4. Theoretical statements analyzing the correctness and exhaustiveness of the learning algorithm with respect to the Psi-Equivalence class.

Strengths: The main strength of this paper is mainly theoretical and methodological. The development of the framework (as outlined above) is meticulous, thorough, and the contributions seem sound and significant for the field of causal discovery. In particular it is appreciated that the authors very thoroughly discussed their method in comparison to existing work in the Appendix, through theoretical statements and clear examples.

Weaknesses: The main weakness of this work is the lack of empirical evaluation, which is the reason this paper is a borderline accept. While the experiment in the appendix is appreciated and reassuring, it would be very useful if the authors could including more extensive experiments, including on real data, with multiple baselines.

Correctness: To the best of my knowledge, the method and the theoretical statements seem correct.

Clarity: The paper is clear and well written given the abundance of information and theoretical statements that it needs to present with the limited space in the main text.

Relation to Prior Work: The prior work is very extensively discussed and related to in the appendix.

Reproducibility: Yes

Additional Feedback: Two minor points: 1. Theorem 4 in the appendix is not referenced anywhere in the text, and is almost the same as Definition 2 in the main text. 2. References in the appendix seem incomplete. All the related work should also be cited in the appendix, since it is so extensively discussed. ******** After author feedback and discussion *************** After carefully considering the rebuttal and other reviewer's comments, and after discussing the paper, I think that although this submission provides detailed comparison to some related work (some of which was not discussed detail in the submission), this comparison should extend to additional comparisons. In particular, in [Zhang et al.] (Section 4.1) it is discussed that invariances are a special case of independent causal mechanisms, and thus it seems to me that it could be potentially restrictive. In addition, the notion of unknown interventional targets is related to not knowing which independent causal mechanisms undergo changes as treated in the above-cited work. Therefore, it would be very beneficial to this already good paper if the authors could possibly include discussion, analysis and if time permits, even empirical experiments comparing to the algorithm based on independent causal mechanisms in the camera-ready version of the manuscript. References: Zhang, K., Huang, B., Zhang, J., Glymour, C., & Schölkopf, B. (2017, August). Causal discovery from nonstationary/heterogeneous data: Skeleton estimation and orientation determination. In IJCAI: Proceedings of the Conference (Vol. 2017, p. 1347). NIH Public Access.


Review 4

Summary and Contributions: This paper introduces a new framework for learning causal graphs based on a combination of observational and interventional data. The authors specifically consider the case of having access to soft interventions where the interventional targets may not be known. Importantly, their framework allows for latent variables, which is not the case for many existing causal inference algorithms. The authors introduce a new property and a corresponding equivalence class (\Psi-markov equivalence class) for characterizing whether two causal graphs and their intervention targets belong to this equivalence class. Subsequently, they introduce an algorithm, \Psi-FCI for learning this equivalence class. They show that the algorithm is sound and complete.

Strengths: The paper addresses, to my knowledge, an important gap in prior work by developing an algorithm that allows for latent variables in causal structure discovery from observational and interventional data. The authors provide an extensive theoretical comparison of their algorithm with previous works through examples in the appendix. The presented equivalence class and algorithm is theoretically well developed. The authors provide important theoretical claims about their algorithm and include examples for further clarification.

Weaknesses: My major point of criticism is the lack of empirical results for the proposed algorithm. Although authors provide some evaluation in the Appendix Fig 6, this part needs to be more expanded. It would be great if authors could show their algorithm’s performance in simulation and on real data as well as how it compares to other algorithms such as those mentioned in prior work. Since the main advantage of this work is the ability to handle latent variables, it would be particularly great to showcase the algorithm in this setting. In addition, given data, it’s not immediately clear how Algorithm 1 would be applied since the input to the algorithm is a tuple of distributions. I think that writing the algorithm in terms of data would be more helpful and useful for the practitioner. The authors should mention specific tests (e.g. F-tests) that would be used for testing invariances. Minor points: How does \Psi-FCI differ from ref [25]. I did not see this discussed in the main paper or the appendix. How does c-faithfilness compare to normal faithfulness? Could the authors add a point on applicability/lack of of their work to hard interventions?

Correctness: The claims appear to be correct, although I did not thoroughly check.

Clarity: The paper is overall well-written.

Relation to Prior Work: The authors make a through theoretical comparison to prior work in the appendix. However, the paper could benefit from empirical comparisons on simulation and real data with prior work.

Reproducibility: Yes

Additional Feedback:

[Author Response · NeurIPS 2020]

We thank the reviewers' time and energy for reading our paper and providing all these comments and suggestions. We are also appreciative for the positive feedback, including that this work has "solid/important theoretical results" (R2/R5), "meticulous, thorough" development of the graphical characterization (R4), and the complete algorithm that "looks inspiring" (R3). Below, we address some of the concerns raised by the reviewers.

**Comparison to (Kocaoglu et al., 2019) [14] (R2).** We provide a detailed comparison of our work to that of [14] in Appendix D.3. Unfortunately, we could not include this material in the main paper due to the lack of space. In short, we consider the problem of learning from interventional data with *unknown interventional targets*, a setting which cannot be handled by the work in [14]. When considering the results in [14], we make the following contributions:

1. We formulate the $\Psi$-Markov property and derive a graphical characterization that subsumes that of [14], which is formally shown in Proposition 7, Appendix D.3.
2. We show that the algorithm introduced in [14] is not applicable under unknown interventional targets and present a complete algorithm ($\Psi$-FCI) under unknown interventional targets (Example 10).
3. We handle causal sufficiency as a special case of the derived results. Subsection 3.1 establishes a graphical characterization for this case, and Section C, in the Appendix, presents an algorithm for learning an equivalence class from interventional data under causal sufficiency. We prove this algorithm to be complete for both known and unknown interventional targets. The work in [14] does not discuss the causally sufficient case nor completeness.

**C-faithfulness versus Faithfulness (R2 & R5).** As noted by R2, c-faithfulness is indeed stronger than the faithfulness assumption, which is common for learning Markov equivalence classes from when only data from one distribution, the observational one is available. A similar assumption is also required in other works [10, 18, 32, 14], but it is phrased slightly differently depending on the specific setting. The preliminary experimental results in Appendix D suggest that c-faithfulness largely holds, especially in the discrete case. However, we do understand the concern of R2 that c-faithfulness may not hold in some settings, in a similar fashion as the faithfulness assumption in the observational case. The present work provides necessary conditions to establish the theoretical limitations of inferring causal invariances from the combination of multiple datasets, and it paves the way for heuristic and approximation algorithms that may weaken the c-faithfulness assumption, akin to weakening faithfulness in the observational case (e.g., Zhalama, Zhang, J., Mayer, W. (2017). Weakening faithfulness: some heuristic causal discovery algorithms. JDCA, 3(2), pp. 93-104).

**Further Empirical Evaluation (R4 & R5).** We agree that additional experiments would be helpful to refine the theoretical understanding under finite samples and other empirical constraints. In this spirit, we have conducted experiments on Sachs data [26] after the submission. We indeed observe discrepancies in the recovered structures compared to JCI [18]. We will provide and discuss these findings in relation with the known ground truth. We also conducted synthetic experiments with two other generating causal graphs (Fig.1(a) and Fig.4(a) with discussions in Ex.6 and Ex.8, respectively) and the results look similar to the ones we included in the paper (Fig.6(a)). For the camera ready, we will conduct more synthetic experiments on random causal graphs and include the results.

**Theorem 4 in the Appendix (R4).** Thm. 4 establishes that a tuple of distributions that is generated by a causal graph satisfies the $\Psi$-Markov property relative to the graph and the true set of interventional targets (lines 174-176). This ascertains that the equivalence class learned by $\Psi$-FCI is non-empty (Theorem 2), i.e., the equivalence class is guaranteed to include at least the generating causal graph and the true unknown set of interventional targets. We will make this point more prominent by referencing the theorem in the main text (instead of only the section, line 176).

**Testing the Distributional Invariances (R5).** There are different ways of implementing hypothesis testing for the distributional invariances discussed in line 22 of $\Psi$-FCI, which can be seen as evaluating statements like $|\hat{P}_i(y|w) - \hat{P}_j(y|w)| \leq \epsilon$, where the hat represents the empirical distribution. $\Psi$-FCI is agnostic to the particular implementation of the test, which is in general chosen based on the specific details of the setting. Still, for concreteness, when the support of two distributions $\hat{P}_i(y, w)$ and $\hat{P}_j(y, w)$ is the same, testing whether $\hat{P}_i(y|w)$ is equal to $\hat{P}_j(y|w)$ can be done as follows. First, define a binary variable $F$ which is set to 0 for a sample from $i$ and set to 1 for a sample from $j$. Then, test if $I(F; Y|W)$ is zero. In our experiments, we use the previous method to test the distributional invariances.

**Hard Interventions (R5).** The presented characterization and algorithm are sound under hard interventions; however, the equivalence class can be further refined in this setting due to the change in the adjacencies of the graphical model following from the do-intervention. To understand the subtlety, consider the graphs $\mathcal{G} = \{X \to Y\}$, $\mathcal{D} = \{X \leftarrow Y\}$ and $\mathcal{I} = \langle \{X\}, \{Y\} \rangle$. The pairs $\langle \mathcal{G}, \mathcal{I} \rangle$ and $\langle \mathcal{D}, \mathcal{I} \rangle$ are $\Psi$-Markov equivalent according to Thm. 1. However, the graphs are distinguishable under hard interventions since $(X \not\perp\!\!\!\perp Y)_{\mathcal{G}_{\overline{X}}}$ while $(X \perp\!\!\!\perp Y)_{\mathcal{D}_{\overline{X}}}$. Given this realization, we opted to avoid discussing hard interventions since the results are somewhat direct but much weaker.

**Comparison to (Rothenhäusler et al., 2015) [25] (R5).** We briefly mentioned this work in the introduction, lines 68-69. We ended up not doing a detailed comparison to $\Psi$-FCI given the very nature of both approaches, they are not really comparable. On the one hand, [25] considers the broader class of cyclic causal models while ours is restricted to acyclic models. On the other hand, our work makes no assumption about the functional form or type of soft intervention, while [25] considers linear causal relations and shift interventions. We will reflect this discussion in the paper.

[Meta-Review · NeurIPS 2020]

This paper is concerned with a line of research which has recently attracted much attention, causal discovery from data with soft interventions with unknown targets. The paper extends previous results and proposes an improve scheme to solve the problem. The paper is nice written, with completeness results of the proposed scheme. Its arguments for using pairwise comparison of the distributions are clear and sensible. Reviewers feel positive about the paper and it is worth accepting; at the same time, they feel that the paper would be stronger if a comparison against independent change-based causal discovery methods, e.g., CD-NOD ([37] and subsequent publications), were given. As illustrated in the paper (supplementary material), if one uses invariance for causal discovery from multiple distributions, then pairwise comparison may be superior to the joint comparison scheme. This 'non-monotonicity' property of invariance also suggests that the condition of invariance may be restrictive. In fact, it is a particular case of the independence change principle [37]--a constant is independent from everything. With independent change principle, one can directly benefit from more domains/distributions. Reviewers are aware of the fact that this paper adopts the FCI framework while CD-NOD uses the PC framework; nevertheless, the authors may want to consider providing a theoretical comparison and/or empirical comparisons on systems with or without confounders.